# Using partial least squares to identify a dietary pattern associated with obesity in a nationally-representative sample of Canadian adults: Results from the Canadian Community Health Survey—Nutrition 2015

**Alena (Praneet) Ng** [1]☯, **Mahsa Jessri**[2,3,4]☯¤, **Mary R. L'Abbe**[1]☯*

**1** Department of Nutritional Sciences, Temerty Faculty of Medicine, University of Toronto, Toronto, Ontario, Canada, **2** Clinical Epidemiology Program, Ottawa Hospital Research Institute, Ottawa, Ontario, Canada, **3** Health Analysis Division, Statistics Canada, Government of Canada, Ottawa, Ontario, Canada, **4** School of Epidemiology and Public Health, University of Ottawa, Ottawa, Ontario, Canada

☯ These authors contributed equally to this work.
¤ Current address: Food, Nutrition and Health Program, Faculty of Land and Food Systems, The University of British Columbia, Vancouver, British Columbia, Canada
* mary.labbe@utoronto.ca

**Data Availability Statement:** Data analyzed in this study were from The Canadian Community Health

## Abstract

### Background

Hybrid methods of dietary patterns analysis have emerged as a unique and informative way to study diet-disease relationships in nutritional epidemiology research.

### Objective

To identify an obesogenic dietary pattern using weighted partial least squares (wPLS) in nationally representative Canadian survey data, and to identify key foods and/or beverages associated with the defined dietary pattern.

### Design

Data from one 24-hr dietary recall data from the cross-sectional Canadian Community Health Survey-Nutrition (CCHS) 2015 ($n = 12,049$) were used. wPLS was used to identify an obesogenic dietary pattern from 40 standardized food and beverage categories using the variables energy density, fibre density, and total fat as outcomes. The association between the derived dietary pattern and likelihood of obesity was examined using weighted multivariate logistic regression. Key dietary components highly associated with the derived pattern were identified.

### Results

Compared to quartile one (i.e. those least adherent to an obesogenic dietary pattern), those in quartile four had 2.40-fold increased odds of being obese (OR = 2.40, 95% CI = 1.91,

Survey – Nutrition 2015 Master Files, which can be accessed through Statistics Canada Research Data Centres. Information and by following the protocol outlined in the Methods section (https://www.statcan.gc.ca/eng/microdata/data-centres).

**Funding:** This research was supported by funds to the Canadian Research Data Centre Network (CRDCN) from the Social Sciences and Humanities Research Council (SSHRC), the Canadian Institute for Health Research (CIHR), the Canadian Foundation for Innovation (CFI) and Statistics Canada. MRL was supported by a research grant from the Burroughs Wellcome Fund Innovation in Regulatory Science Award. MJ was supported by the Canadian Institutes of Health Research (CIHR) Banting Fellowship, a CIHR Fellowship and Banting Discovery Award. APN was supported by the Canadian Institutes of Health Research (CIHR) Master's Fellowship. The funders had no role in study design, data collection and analysis, decision to publish, or preparation of the manuscript.

**Competing interests:** The authors have declared that no competing interests exist.

**Abbreviations:** AMPM, Automated Multiple Pass Method; BMI, body mass index; BNS, Bureau of Nutritional Sciences; CCHS, Canadian Community Health Survey-Nutrition; CI, confidence interval; DGA, Dietary Guidelines for Americans; DGAI, Dietary Guidelines for Americans Adherence Index; EER, estimated energy requirement; IOM, Institute of Medicine; NCDs, non-communicable diseases; OR, odds ratio; PCA, principal component analysis; PLS, partial least squares; Q, quartile; RRR, reduced rank regression; SDS, simplified dietary pattern score; WHO, World Health Organization; wPLS, weighted partial least squares.

3.02, P-trend< 0.0001) with a monotonically increasing trend. Using a factor loading significance cut-off of $\geq|0.17|$, three food/beverage categories loaded positively for the derived obesogenic dietary pattern: fast food (+0.32), carbonated drinks (including energy drinks, sports drinks and vitamin water) (+0.30), and salty snacks (+0.19). Seven categories loaded negatively (i.e. in the protective direction): whole fruits (-0.40), orange vegetables (-0.32), "other" vegetables (-0.32), whole grains (-0.26), dark green vegetables (-0.22), legumes and soy (-0.18) and pasta and rice (-0.17).

## Conclusion

This is the first study to apply weighted partial least squares to CCHS 2015 data to derive a dietary pattern associated with obesity. The results from this study pinpoint key dietary components that are associated with obesity and consumed among a nationally representative sample of Canadians adults.

## Introduction

Although mortality rates are decreasing in Canada, the prevalence and burden of non-communicable diseases (NCDs) remains high [1, 2]. One of the estimated top risk factors contributing to the increased prevalence of NCDs among Canadians is high body mass index (BMI) [2]. While self-reported obesity rates have not increased in the past decade, with 6 in 10 Canadian adults either overweight or obese [3], multi-comorbidity continues to be a problem [1]. Regarding the overall health of Canadians and years lived free from the burden of high BMI and NCDs, the picture needs to be drastically improved.

Fortunately, many of the risk factors which affect obesity and NCD risk are behavioural. One of the leading risk factors is poor diet, with dietary risks such as low consumption of fruits and vegetables and high consumption of processed meats contributing to the risk of obesity and NCDs [2, 4]. As overweight and obesity tend to be on the pathway to chronic disease, it is important to understand the nuanced and complex relationship between diet and weight management.

A dietary patterns approach–in which the intake and quantities of dietary components (foods, nutrients and beverages) are studied together for their association with a health outcome within a population–is one way to study diet-disease relationships. "Hybrid" methods of dietary patterns analysis, such as reduced rank regression (RRR) and partial least squares (PLS), are data reduction techniques which use response or "training" variables on the pathway to disease in order to identify a dietary pattern from the data which accounts for the most variation in both predictor and response variables. Both RRR and PLS methods can identify key dietary components which load highly in the dietary pattern of interest, i.e. components which are the most highly associated with a disease outcome of interest.

A weighted PLS model was recently used to identify an obesogenic dietary pattern within a nationally-representative sample of Canadian adults from the cross-sectional Canadian Community Health Survey (CCHS) 2004 [5]. Respondents' higher adherence to the obesogenic pattern (defined as energy-dense, high fat and low fibre) was associated with significantly increased odds of being obese, with or without accompanying chronic disease. The wPLS model was also able to identify key dietary components associated with obesity, the top three

being: fast food, carbonated drinks and refined grains in the positive direction; and whole fruits, dark green vegetables, "other" vegetables and vegetable juices in a protective direction.

CCHS 2015, the most-current nationally-representative data available on Canadians' dietary intakes, was released in June 2017. To our knowledge, examining dietary patterns through PLS and their associations with obesity have not been conducted on this dataset. The objectives of this study were to therefore: a) identify an obesogenic dietary pattern using weighted PLS, b) to examine the associations between this dietary pattern and obesity, and c) to compare the results to earlier findings using CCHS 2004 data.

## Subjects and methods

### Study population and data collection

Data from the Canadian Community Health Survey-Nutrition (CCHS) 2015 were used for this study. CCHS 2015 is the most recent, nationally-representative cross-sectional survey data available which provides information on the dietary intakes of Canadians, with a sampling frame designed to represent >98% of the Canadian population. The survey excludes those living in the territories, Armed Forces, prison or long-term care facilities. All data were collected under the authority of the Statistics Act of Canada.

The original sample size of CCHS 2015 was 20,487. Weighted partial least squares analyses were originally performed on a sample of 13,598 adults to characterize the dietary patterns of the adult Canadian population. This sample of adults consisted of 50% males and 50% females. Of the sample, 17% were between 19-30y, 38% were between 31-50y, and 45% were above 50y. The percentage of normal weight individuals, after correcting for missing measured height and weight, was 35%; 37% of respondents were overweight, and 28% were obese. We then further removed pregnant and/or breastfeeding women, those who were underweight (BMI<18.5), and those who were missing measures for dietary energy intake, height, weight, physical activity, smoking status, education or marital status for a sample size of $n$ = 12,049. For regression analyses of the association between dietary patterns and obesity with or without chronic disease, a further 304 respondents were removed due to missing self-reported chronic disease status for an analytical sample size of $n$ = 11,745 for these secondary analyses.

### Dietary assessment

Respondents in CCHS 2015 were randomly chosen per household to complete an in-home interview assisted by trained personnel; the interview comprised of a general health questionnaire for general sociodemographic and lifestyle information, one 24-hour dietary recall, and the measurement of height and weight (with respondent consent).

All individuals were guided through the 24-hr dietary recall using a computerized modification of the USDA's automated multiple-pass method (AMPM) [6]. Respondents were instructed to recall all foods, beverages and multivitamins consumed over the past 24 hours. After the initial recall and interview, approximately 30% of respondents were randomly selected for a second 24-hr recall seven-to-ten days later over the phone. Foods, meals and recipes reported in each 24-hr dietary recall were disaggregated into their corresponding nutritional composition using the 2015 Canadian Nutrient File [7].

### Dietary patterns analysis

**A priori *method*: *2015 DGAI (Dietary Guidelines for Americans Adherence Index)*.** The 2015 Dietary Guidelines for Americans Adherence Index (DGAI) is a diet quality index which measures adherence to the 12 energy-based Healthy Eating Patterns found in the 2015–2020

Dietary Guidelines for Americans (DGA) [8–10]. The 12 Healthy Eating Patterns were created for the 2015–2020 DGA after extensive systematic review of the literature on dietary patterns for chronic disease prevention up to 2012 [4]. Recently, the 2015 DGAI has been applied to a sample of Canadian adults from CCHS 2004 and has been shown to be associated with improved diet quality and a lower likelihood of obesity in this sample [5, 11]. Validity and reliability of the 2015 DGAI for use in studying the association of diet with chronic disease was also examined and confirmed previously [5, 11]. For these reasons, the 2015 DGAI was used in this study as the *a priori* dietary pattern method of choice for comparison with newer, more statistically-sophisticated hybrid methods of dietary patterns analysis. This also allows comparison between results from this study–on CCHS 2015 –with similar published results on CCHS 2004 [5].

Further detail on the scoring criteria for the 2015 DGAI can be found in S1 Table.

**Hybrid method: wPLS (weighted partial least squares).** Weighted partial least squares was used to identify a dietary pattern in CCHS 2015 associated with obesity [5, 12]. Partial least squares (PLS) regression is a data-reduction method aimed at explaining the covariance between predictor variables on the pathway to disease and disease-specific responses. Unlike other data-reduction methods, such as factor analysis or principal component analysis (PCA), PLS uses the inclusion of response variables to "train" the multivariate model to produce a dietary pattern of interest, given a set of dietary predictor variables. PLS aims to produce factors (or dietary patterns) which account for the most variation in both predictor and response variables.

To define an "obesogenic" dietary pattern, the three response variables energy density, total fat intake and fiber density were used. These variables have been previously used in the literature in both reduced rank regression (RRR) and PLS models to derive dietary patterns associated with obesity [5, 13, 14], and are based on the World Health Organization (WHO) recommendations for dietary interventions to reduce the risk of chronic disease [15]. As an additional sensitivity test, we also examined the wPLS-derived dietary pattern when the response variable total fat was exchanged for saturated fat. The total fat model out-performed the saturated fat model to explain more variation in the data (S2 Table); we therefore chose the original model with energy density, total fat intake and fiber density as response variables for all analyses.

For easier interpretation, all foods were aggregated into larger food categories using the Bureau of Nutritional Sciences (BNS) Food Codes. Forty food variables were chosen as predictor variables in the wPLS model and can be found in S3 Table. These predictor variables differ slightly from those in previous research on CCHS 2004 data in that they include new food categories which have been added to CCHS 2015 [5, 16].

wPLS dietary pattern scores were outputted as the product of food intake and factor loading in the wPLS model, summed across all forty predictor variables. To validate the derived dietary pattern from wPLS, random cross-split validation was performed five times. This method splits the sample randomly into half and performs the wPLS again on the five half-samples. The Pearson correlations between the identified wPLS patterns for each half-split and the original sample was then examined. The mean Pearson correlation coefficient between the continuous wPLS scores derived from the full dataset and the five half-split datasets was r = 0.9986, P<0.0001.

**Simplified dietary pattern score (SDS).** Using the methodology of Schulze et al., a "simplified" diet score (SDS) was constructed from the results of the weighted PLS regression [17]. Only those response variables most highly associated with the obesogenic dietary pattern from wPLS (using a factor loading of ≥|0.17| to denote significance) were kept and their standardized intakes summed for a final SDS. These dietary components were: fast food, carbonated

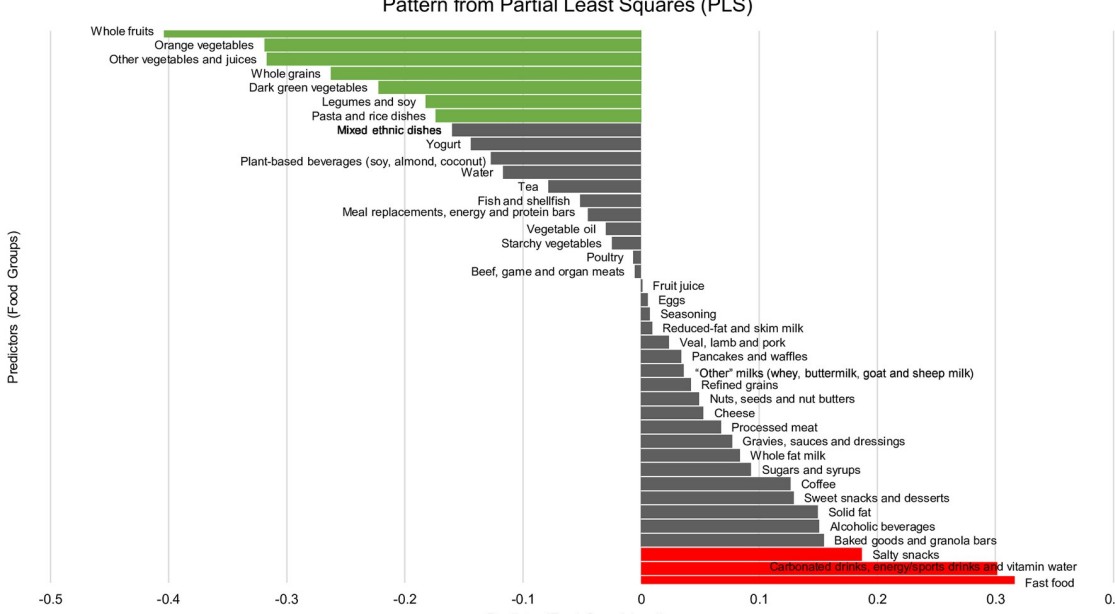

**Fig 1. Predictor loadings for the energy-dense, high-fat and low-fiber density ("obesogenic") dietary pattern derived from a weighted partial least squares (PLS) analysis in adult participants of the Canadian Community Health Survey 2015 (*n* = 13,598).**

drinks (including energy drinks, sports drinks and vitamin water) and salty snacks (weighted to 1); and whole fruits, orange vegetables, "other" vegetables (including vegetable juice), whole grains, dark green vegetables, legumes and soy and pasta and rice (weighted to -1) (Fig 1). A higher SDS score represents higher adherence to an obesogenic dietary pattern, after omitting those foods which loaded insignificantly in the wPLS model. The SDS may be more beneficial than scores derived from the total wPLS model in that it may be more easily translatable across datasets and population sub-samples, while preserving the integrity of the derived dietary pattern by retaining only those highly-loaded food predictors [5].

**Accounting for dietary energy misreporting.** To account for energy misreporting, respondents were classified as under-reporters, over-reporters and plausible reporters of energy intake by comparing their reported energy intakes from their 24-hr dietary recall to their estimated energy requirements (EER) based on the Institute of Medicine (IOM)'s EER equations [18]. Respondents were considered under-reporters if their reported energy intake was <70% of their EER; over-reporters if their reported energy intake was >142% of their EER; and plausible reporters of energy intake if their reported energy fell within 70–142% of their EER [19].

**Statistical analyses.** All analyses were performed at Statistics Canada's Research Data Center in Toronto, Ontario using SAS Version 9.4 and JMP Genomics 11.2 (SAS Institute). Sample survey weights were provided by Statistics Canada and applied to all analyses to ensure population-level estimates. Bootstrapping with balanced repeated replication (500 replicates) was performed to ensure reliable standard errors and coefficients of variation from complex survey data.

Pearson correlations were examined between continuous scores for: 1) the obesogenic dietary pattern generated using wPLS and 2) the SDS, compared with the wPLS predictor variables, wPLS response variables, and dietary components of the 2015 DGAI. The relationship

between the two scores were also tested categorically for their association with intake of specific predictor food variables, nutrient intake, and lifestyle and sociodemographic variables. To do this, continuous scores for both wPLS and SDS were split into quartiles, and least squares means or percentages calculated where appropriate. Intakes in Table 4 were adjusted for age, sex, energy and misreporting status; intakes in Table 5 were adjusted for age, sex and misreporting status with intakes expressed per 1000kcal or as a percentage of total energy where appropriate.

To examine the relationship between dietary pattern scores with obesity, weighted multivariable logistic regression was performed with quartile categories of either the wPLS or SDS scores as independent variables and obesity (BMI≥30) as the dependent variable, adjusting for age, sex, physical activity, smoking status and misreporting status [5]. To investigate the possibility of different obesity phenotypes (i.e. "healthy" obesity and "unhealthy" obesity) [5, 20] and their association with dietary patterns, obesity analyses were also performed with a logistic regression model with a four-category, multivariate outcome variable [the four categories were: normal weight without chronic conditions (reference), normal weight with ≥1 chronic conditions, obesity (BMI≥30) without chronic conditions, (i.e. "healthy" obesity) and obesity with ≥1 chronic conditions (i.e. "unhealthy obesity")]. The presence of ≥1 chronic conditions was identified by respondents reporting one or more of either hypertension, cardiovascular diseases, cancer or diabetes in the general health questionnaire portion of CCHS 2015.

Statistical significance for all analyses were set at P<0.0001.

## Results

Table 1 demonstrates sociodemographic and lifestyle characteristics of the sample of Canadian adults from CCHS 2015 across quartile categories of the obesogenic dietary pattern score (wPLS score) and simplified dietary pattern score. Moving from Q1 to Q4, respondents were more likely to be unmarried, younger, have a higher BMI, and smoke daily (P-trend<0.0001). In terms of adherence to physical activity guidelines, there was no significant trend with the wPLS score (P-trend = 0.1416), and only a weak association with the SDS (P-trend = 0.0398).

### Identification of dietary patterns from CCHS 2015

Three dietary patterns (i.e. factors) were identified through wPLS. The first pattern was kept for analysis as it explained the most variation in both the response variables (energy density, total fat intake and fiber density) and the predictor variables (40 BNS food categories). This pattern loaded positively for energy density and total fat intake and negatively for fibre density–it therefore represents an obesogenic dietary pattern (defined as energy-dense, high fat, and low fibre) in this sample of CCHS 2015 adults.

Fig 1 illustrates the 40 BNS food/beverage category predictors and their respective factor loadings from the obesogenic dietary pattern generated from wPLS. The factor loadings for each BNS food/beverage category represents the contribution of each category to the dietary pattern. Using a factor loading cut-off of ≥|0.17| [5, 17], intake of three BNS food/beverage categories loaded positively for this dietary pattern: fast food (+0.32), carbonated drinks (including energy drinks, sports drinks and vitamin water) (+0.30), and salty snacks (+0.19). Seven BNS food/beverage categories loaded negatively: consumption of whole fruits (-0.40), orange vegetables (-0.32), "other" vegetables (including vegetable juice) (-0.32), whole grains(-0.26), dark green vegetables (-0.22), legumes and soy (-0.18) and pasta and rice (-0.17).

Table 2 depicts Pearson correlations between the BNS food/beverage predictor variables which loaded significantly in the wPLS model (using a loading factor cut-off of ≥|0.17|) and:

**Table 1. Analysis of sociodemographic and lifestyle characteristics across quartile categories of the energy-dense, high-fat and low-fiber density ("obesogenic") dietary pattern score generated from wPLS, and the simplified dietary pattern score (SDS) among Canadian adults from CCHS-Nutrition 2015 (*n* = 12,049).**

| | wPLS quartiles | | | | SDS quartiles | | | |
|---|---|---|---|---|---|---|---|---|
| | **1 (Healthiest)** | **2** | **3** | **4 (Least Healthy)** | **1 (Healthiest)** | **2** | **3** | **4 (Least Healthy)** |
| Total DGAI score | 10.7±0.1 | 9.4±0.1 | 8.4±0.1 | 7.5±0.1 | 10.5±0.1 | 9.3±0.1 | 8.4±0.1 | 7.7±0.1 |
| "Food Intake" subscore | 4.8±0.1 | 4.0±0.1 | 3.5±0.1 | 3.4±0.1 | 4.9±0 | 4.1±0.1 | 3.5±0.1 | 3.2±0 |
| "Healthy Choices" subscore | 5.8±0 | 5.4±0 | 4.9±0 | 4.1±0 | 5.6±0 | 5.2±0.1 | 4.9±0.1 | 4.5±0.1 |
| Female, % | 54.2±1.5 | 56.6±1.5 | 33±1.5 | 33±1.5[2] | 50.6±1.6 | 53.5±1.6 | 54.9±1.6 | 39.2±1.7 |
| Age, years | 50±0.5 | 51±0.5 | 48.9±0.5 | 45.9±0.6 | 49±0.5 | 50.4±0.6 | 50.6±0.6 | 45.8±0.6 |
| BMI, kg/m$^2$ | 27.1±0.2 | 27.7±0.2 | 28.2±0.2 | 28.7±0.2 | 27.2±0.2 | 27.4±0.2 | 28.2±0.2 | 28.9±0.2 |
| Obesity, % | 23.3±1.2 | 27.2±1.2 | 29.4±1.7 | 33.6±1.7 | 23.6±1.2 | 24.9±1.2 | 30.9±1.6 | 34.5±1.8 |
| Current daily smokers, % | 7.3±0.8 | 10.4±1.1 | 14.3±1.2 | 22.4±1.7 | 7.6±0.9 | 9.8±0.9 | 17±1.5 | 20.1±1.5 |
| Met physical activity guidelines[1], % | 47.6±1.7 | 44.7±1.9 | 41.5±1.8 | 43.2±2.1[3] | 46.8±1.8 | 44.2±1.8 | 42.4±1.8 | 43.7±1.9[4] |
| Highest household education, % | | | | | | | | |
| < Secondary school | 4.7±0.4 | 6.1±0.5 | 7.9±0.6 | 10.5±0.8 | 4.7±0.4 | 6.1±0.5 | 8.2±0.6 | 10.3±0.8 |
| Post-secondary education | 52.5±1.8 | 45.3±1.8 | 38.6±1.8 | 31.4±1.6 | 52.1±1.8 | 45.4±1.8 | 37.6±1.6 | 31.9±1.8 |
| Immigrant, % | 42.2±1.8 | 31.2±1.9 | 21.2±1.6 | 13.2±1.5 | 39.7±1.8 | 33.2±2 | 22.8±1.6 | 11.4±1.2 |
| Marital status, % | | | | | | | | |
| Single | 16.7±1.1 | 19.6±1.3 | 23.3±1.5 | 23±1.5 | 17±1.1 | 19.7±1.4 | 22.2±1.4 | 23.8±1.6 |
| Married | 70.3±1.6 | 66.1±1.8 | 61±1.8 | 61.4±1.9 | 69.8±1.6 | 65.9±1.8 | 62.5±1.8 | 60.4±2 |

CCHS: Canadian Community Health Survey; SDS: simplified dietary pattern score; wPLS: weighted partial least squares. Values are means or percentages ± standard errors. Estimates are weighted LS means or percentages from a regression model adjusted for age and sex with bootstrapping to ensure accurate standard errors when using survey data. P-trends were estimated with the use of the ED, HF, LFD dietary pattern score or the simplified dietary pattern score in their continuous form and represents the P-value associated with the linear regression coefficient.

All P-trends were <0.0001 unless otherwise noted.

[1]Current Canadian physical activity guidelines for adults states reaching a goal of 150 minutes of moderate/vigorous-intensity physical activity per week.

[2]P-trend = 0.0002.

[3]Ptrend = 0.1416.

[4]P-trend = 0.0398.

A) response variables in the wPLS model (i.e. energy density, total fat and fibre density; leftmost columns); and B) dietary pattern scores (right-most columns). As expected, the wPLS scores and SDS were highly correlated (*r* = 0.87, P<0.0001), and both scores showed moderate-to-high inverse correlations with the 2015 DGAI (-0.59 and -0.5, respectively; P<0.0001). All response variables (energy density, total fat intake and fiber density) were also moderately-to-highly correlated with the both SDS and wPLS scores and in the expected direction (SDS and wPLS; lowest *r* = 0.19 for total fat and SDS, highest *r* = 0.7 for energy density and wPLS).

Table 3 demonstrates Pearson correlations between the sub-components of the 2015 DGAI and both dietary pattern scores from wPLS and SDS. Overall, both scores exhibited significant inverse correlations between the "Food Intake" subscore and the "Healthy Choices" subscore of the 2015 DGAI. Correlation coefficients between the "Food Intake" sub-components of the 2015 DGAI and both dietary pattern scores were generally low-to-moderate in the inverse direction, with the largest correlation between "variety of vegetable and whole fruit intake" and SDS score (*r* = -0.43, P<0.0001). Results were similar for the "Healthy Choices" sub-components of the DGAI, with a significantly large inverse correlation between the fiber density sub-component and both dietary pattern scores (*r* = -0.64 between fiber density and wPLS and *r* = -0.51 between fiber density and SDS).

**Table 2. Pearson correlation coefficients between food predictors[1] which loaded significantly in the wPLS model (i.e. loading factor ≥|0.17|); response variables in the energy-dense, high-fat and low-fiber density ("obesogenic") dietary pattern generated from the wPLS model; the simplified dietary pattern score (SDS); and total scores for the 2015 DGAI among Canadian adults from CCHS-Nutrition 2015 (*n* = 12,049).**

| | Response variables in the wPLS model | | | Dietary pattern scores | | |
|---|---|---|---|---|---|---|
| | **Energy density** | **Fiber density** | **Total energy from fat** | **wPLS** | **SDS** | **2015 DGAI** |
| Predictor variables | | | | | | |
| Positive association | | | | | | |
| Fast food | 0.24 | -0.2 | 0.1 | 0.42 | 0.48 | -0.22 |
| Salty snacks | 0.21 | -0.07 | 0.11 | 0.24 | 0.31 | -0.07 |
| Carbonated drinks | 0.22 | -0.22 | -0.02[6] | 0.4 | 0.43 | -0.22 |
| Inverse association | | | | | | |
| Pasta and rice | -0.16 | -0.02[2] | -0.18 | -0.23 | -0.43 | 0.08 |
| Legumes and soy | -0.08 | 0.22 | -0.02[3] | -0.25 | -0.37 | 0.15 |
| Whole grain breads and cereals | -0.15 | 0.31 | -0.1 | -0.35 | -0.37 | 0.24 |
| Dark green vegetables | -0.22 | 0.19 | 0.02[4] | -0.29 | 0.37 | 0.2 |
| Orange vegetables | -0.3 | 0.22 | -0.08 | -0.42 | -0.48 | 0.16 |
| "Other" vegetables | -0.32 | 0.27 | -0.02[5] | -0.42 | -0.42 | 0.36 |
| Whole fruits | -0.43 | 0.37 | -0.17 | -0.54 | -0.44 | 0.37 |
| Response variables | | | | | | |
| Energy density | 1.00 | -0.47 | 0.46 | 0.7 | 0.57 | -0.52 |
| Fiber density | -0.47 | 1.00 | -0.28 | -0.5 | -0.62 | 0.58 |
| Total energy from fat | 0.46 | -0.28 | 1.00 | 0.3 | 0.19 | -0.27 |
| Dietary pattern scores | | | | | | |
| wPLS | 0.7 | -0.62 | 0.3 | 1.00 | 0.87 | -0.59 |
| SDS | 0.57 | -0.5 | 0.19 | 0.87 | 1.00 | -0.5 |

CCHS: Canadian Community Health Survey; DGAI: Dietary Guidelines for Americans Adherence Index; SDS: simplified dietary pattern score; wPLS: weighted partial least squares.

All P-values were <0.0001 unless otherwise noted.

[1]Detailed description of the foods included for each predictor variable can be found in S3 Table.

[2]P = 0.0128.

[3]P = 0.0528.

[4]0.0305.

[5]P = 0.0149.

[6]0.0324.

## Association between dietary pattern scores and nutritional and sociodemographic profiles in CCHS 2015

Table 4 depicts mean daily intake of the food/beverage predictors from the wPLS model which loaded significantly (i.e. loading factor ≥|0.17|) across quartile categories of the wPLS score and SDS adjusting for age, sex, energy intake and energy misreporting status. Moving from Q1 to Q4 (i.e. higher adherence to an obesogenic dietary pattern), respondents from this sample were more likely to consume the predictor foods/beverages which loaded positively in the wPLS model (fast foods, salty snacks and carbonated drinks) and consume less of the predictor foods which loaded negatively (e.g. whole fruits and whole grains) (P-trend<0.0001). In some cases, intakes were drastically different in Q4 compared to Q1, with an almost 10-fold difference in the mean daily intake of whole fruits between Q4 and Q1 of wPLS score.

Table 5 demonstrates mean daily intake of macro- and micronutrients across quartile categories of the wPLS score and SDS adjusting for age, sex and misreporting status. As expected,

**Table 3. Pearson correlation coefficients between components of the 2015 DGAI[1]; the energy-dense, high-fat and low-fiber density ("obesogenic") dietary pattern score generated from wPLS; and the simplified dietary pattern score (SDS) among Canadian adults from CCHS-Nutrition 2015 (*n* = 12,049).**

| | *A priori* dietary pattern score (i.e. 2015 DGAI) | | | Hybrid dietary pattern scores | |
|---|---|---|---|---|---|
| | **Total 2015 DGAI score** | **"Food Intake" subscore** | **"Healthy Choices" subscore** | **wPLS** | **SDS** |
| DGAI "Food Intake" subscore | | | | | |
| Vegetables and legumes | 0.65 | 0.85 | 0.05 | -0.31 | -0.35 |
| Whole fruits | 0.41 | 0.39 | 0.2 | -0.41 | -0.36 |
| Variety of fruits and vegetables | 0.72 | 0.89 | 0.12 | -0.42 | -0.43 |
| Grains | 0.1 | 0.16 | -0.01 | 0.06 | 0[7] |
| Meat and beans | 0.18 | 0.35 | -0.01 | 0.03[5] | -0.02[6] |
| Dairy | 0.09 | 0.27 | -0.15 | 0.09 | 0.04 |
| Added sugar | 0.24 | 0.39 | -0.05 | -0.22 | -0.19 |
| Total "Food Intake" subscore | -0.15 | 1.00 | 0.03[8] | -0.38 | -0.42 |
| DGAI "Healthy Choices" subscore | | | | | |
| Whole grains | 0.39 | 0.11 | 0.45 | -0.23 | -0.2 |
| Fiber density | 0.67 | 0.33 | 0.64 | -0.64 | -0.51 |
| Total fat | 0.32 | 0.02[2] | 0.46 | -0.19 | -0.11 |
| Saturated fat | 0.39 | 0.01[3] | 0.57 | -0.28 | -0.18 |
| Cholesterol | 0.29 | -0.12 | 0.56 | -0.15 | -0.02[4] |
| Low-fat dairy products | 0.22 | 0.09 | 0.22 | -0.07 | -0.04 |
| Low-fat meat products | 0.3 | 0.23 | 0.2 | -0.1 | -0.14 |
| Sodium | 0.24 | -0.19 | 0.57 | -0.22 | -0.1 |
| Alcohol | 0.18 | -0.07 | 0.34 | -0.14 | -0.07 |
| Total "Healthy Choices" subscore | 0.7 | 0.03[8] | 1.00 | -0.48 | -0.3 |
| Hybrid dietary pattern scores | | | | | |
| wPLS | -0.5 | -0.38 | -0.48 | 1.00 | 0.87 |
| SDS | -0.5 | -0.42 | -0.3 | 0.87 | 1.00 |

CCHS: Canadian Community Health Survey; DGAI: Dietary Guidelines for Americans Adherence Index; SDS: simplified dietary pattern score; wPLS: weighted partial least squares.

All P-values were <0.0001 unless otherwise noted.

[1]Detailed scoring criteria for the 2015 DGAI can be found in S1 Table.

[2]P = 0.0474.

[3]P = 0.338.

[4]P = 0.0432.

[5]P = 0.0003.

[6]P = 0.0083.

[7]P = 0.9522.

[8]P = 0.0003.

energy density, total fat intake and fiber density were significantly different between those in Q4 (i.e. most adherent to an obesogenic dietary pattern) and Q1 (P-trend<0.0001). Compared to Q1, those in Q4 were also more likely to consume more of their daily total energy as saturated fat and alcohol, and consume significantly less beneficial micronutrients such as vitamin C, folate and potassium (P-trend<0.0001 for all).

## Association between dietary pattern scores and obesity in CCHS 2015

Fig 2 illustrates the multivariate-adjusted odds ratios (ORs) and 95% confidence intervals (CIs) for the likelihood of being obese (BMI≥30) across quartile categories of the wPLS score

**Table 4. Weighted mean daily intake of food predictors[1] which loaded significantly (i.e. loading factor ≥|0.17|) across quartile categories of energy-dense, high-fat and low-fiber density ("obesogenic") dietary pattern score generated from wPLS, and the simplified dietary pattern score (SDS) among Canadian adults from CCHS-Nutrition 2015 ($n$ = 12,049).**

| Predictor variables (g/day) | wPLS score quartiles | | | | SDS quartiles | | | |
|---|---|---|---|---|---|---|---|---|
| | 1 (Healthiest) | 2 | 3 | 4 (Least Healthy) | 1 (Healthiest) | 2 | 3 | 4 (Least Healthy) |
| Positive association | | | | | | | | |
| Fast food | 55.1±6.3 | 99.5±7.2 | 123.1±7.7 | 220.5±10.1 | 42.3±6.3 | 87.0±7.2 | 125.4±9.9 | 264.3±8.8 |
| Salty snacks | 6.3±1.6 | 9.0±1.7 | 8.9±1.6 | 20.1±2.7 | 5.1±1.4 | 7.4±1.6 | 9.8±1.6 | 24.2±2.8 |
| Carbonated drinks | 40.8±10 | 82.8±10.9 | 132.4±11.9 | 317.1±22.3 | 34.8±8.2 | 73.4±9.0 | 117.0±11.5 | 379.9±20.1 |
| Inverse association | | | | | | | | |
| Pasta and rice | 145±5.8 | 91.9±5.3 | 51±3.7 | 16±4 | 120.8±5.7 | 94.5±5.8 | 72.9±4.5 | 24.5±4.6 |
| Legumes and soy | 32.8±3.2 | 8.6±1.7 | 3.5±1.4 | 0.1±1.5 | 29.7±3.3 | 10.4±1.8 | 5.9±1.5 | 0.4±1.6 |
| Whole grain breads and cereals | 63.9±3.7 | 33.2±2.5 | 20.5±1.9 | 6.7±2.1 | 66.7±3.6 | 28.8±2.1 | 16.7±1.8 | 10.9±2.0 |
| Dark green vegetables | 41±2.5 | 20.5±1.8 | 13.7±1.6 | 6.9±1.7 | 46.9±2.7 | 18.5±1.6 | 8.9±1.3 | 5.9±1.6 |
| Orange vegetables | 104.3±4.4 | 53.2±3.8 | 31.6±3.0 | 3.6±2.8 | 106.9±4.8 | 49.5±3.4 | 25.8±2.9 | 7.7±2.6 |
| "Other" vegetables | 131.3±5.8 | 65.7±3.6 | 43.1±3.1 | 11.7±3.2 | 128.1±5.9 | 66.1±4.0 | 37.3±3.0 | 17.7±3.0 |
| Whole fruits | 295.0±8.7 | 159.9±7.3 | 86.2±6.7 | 29.2±7.4 | 258.1±8.9 | 160.7±8.1 | 102.1±8.4 | 49.9±5.8 |

CCHS: Canadian Community Health Survey; SDS: simplified dietary pattern score; wPLS: weighted partial least squares. Values (grams per day) are means ± standard errors. Estimates are weighted LS means from a linear regression adjusted for age, sex, energy intake and energy misreporting with bootstrapping to ensure accurate standard errors when using survey data. P-trends were estimated with the use of the ED, HF, LFD dietary pattern score or the simplified dietary pattern score in their continuous form and represents the P-value associated with the linear regression coefficient.

All P-trends were <0.0001.

[1]Based on the 40 food predictors used as exposure variables in the weighted PLS model; see S3 Table for more detail.

and SDS among this sample of Canadian adults. Compared to Q1 (i.e. those least adherent to an obesogenic dietary pattern), those in Q4 had a 2.40-fold increased odds of being obese (OR = 2.40, 95% CI = 1.91, 3.02, P-trend<0.0001), with a trend monotonically increasing in nature. Results for quartile categories of the SDS were similar, with those in Q4 of the SDS having a 91% increased likelihood of being obese compared to Q1 (OR = 1.91, 95% CI = 1.50, 2.44, P-trend<0.0001).

Fig 3 illustrates the multivariate-adjusted odds ratios (ORs) and 95% confidence intervals (CIs) for joint classifications between obesity and chronic disease status (i.e. having self-reported ≥1 chronic conditions) across quartile categories of the wPLS score and SDS among this sample of Canadian adults.

Compared to Q1 (i.e. those least adherent to an obesogenic dietary pattern), those in Q4 had an approximately 2.40-fold increased odds of both "unhealthy" obesity (i.e. obesity with ≥1 chronic conditions; OR = 2.44; 95% CI = 1.33,1.55; P-trend<0.0001) and "healthy" obesity (i.e. obesity with no chronic conditions; OR = 2.45; 95% CI = 1.18,1.65; P-trend<0.001). The odds of being normal weight with chronic conditions was not significant (OR = 1.06, 95% CI = 0.97, 1.03, P-trend = 0.0543). Results for obesity phenotypes across quartile categories of SDS were similar.

A table of exact ORs, CIs and accompanying P-trends for Figs 2 and 3 can be found in S4 and S5 Tables.

## Discussion

To our knowledge, this is the first study to apply weighted partial least squares (wPLS) to CCHS 2015 data to derive a dietary pattern associated with obesity. The results from this study indicate that greater adherence to wPLS-generated dietary pattern defined as energy-dense,

**Table 5. Weighted mean daily intake of macro- and micronutrients across quartile categories of energy-dense, high-fat and low-fiber density ("obesogenic") dietary pattern score generated from wPLS, and the simplified dietary pattern score (SDS) among Canadian adults from CCHS-Nutrition 2015 (n = 12,049).**

| | wPLS score quartiles | | | | SDS quartiles | | | |
|---|---|---|---|---|---|---|---|---|
| | 1 (Healthiest) | 2 | 3 | 4 (Least Healthy) | 1 (Healthiest) | 2 | 3 | 4 (Least Healthy) |
| Energy intake, kcal | 2216±19 | 2189±20 | 2227±18 | 2455±23 | 2266±18 | 2244±19 | 2230±23 | 2393±22 |
| Energy density | 1.3±0 | 1.6±0 | 2±0 | 2.4±0 | 1.4±0 | 1.7±0 | 2±20 | 2.3±0 |
| Carbohydrate, % energy | 50.8±0.4 | 47.8±0.4 | 44.9±0.4 | 44±0.4 | 49.7±0.4 | 46.9±0.4 | 44.3±0.4 | 45.9±0.4 |
| Fiber density, g/1000kcal | 13.7±0.2 | 10.1±0.1 | 7.6±0.1 | 6.5±0.1 | 12.9±0.2 | 9.7±0.2 | 7.6±0.1 | 6.8±0.1 |
| Total fat, % energy | 29.1±0.3 | 31.9±0.3 | 34.1±0.4 | 36.2±0.4 | 30.3±0.3 | 32.6±0.3 | 34.3±0.4 | 34.9±0.4 |
| SFA, % energy | 8.7±0.1 | 10.4±0.1 | 11.5±0.2 | 12±0.2 | 9.3±0.1 | 10.7±0.1 | 11.4±0.2 | 11.5±0.2 |
| MUFA, % energy | 10.9±0.2 | 11.8±0.2 | 12.6±0.2 | 13.6±0.2 | 11.3±0.2 | 12.2±0.2 | 12.6±0.2 | 13.2±0.2 |
| PUFA, % energy | 6.7±0.1 | 6.8±0.1 | 7±0.2 | 7.4±0.2[7] | 6.8±0.1 | 6.7±0.1 | 7.3±0.2 | 7.1±0.1[1] |
| Protein, % energy | 17.6±0.2 | 17.1±0.2 | 16.6±0.2 | 15.8±0.2[8] | 17.4±0.2 | 16.9±0.2 | 16. ±0.2 | 15.6±0.2 |
| Alcohol, % energy | 2.4±0.2 | 3.3±0.2 | 4.5±0.3 | 4.1±0.3 | 2.6±0.2 | 3.6±0.3 | 4.5±0.3 | 3.6±0.3 |
| Cholesterol density, mg/1000kcal | 135.3±5.4 | 149.4±4.8 | 149.9±4.9 | 147.6±4.9 | 140±4.9 | 151.7±4.7 | 156.3±4.8 | 134.7±5.1[2] |
| Calcium density, mg/1000kcal | 445.8±7.1 | 425±7.9 | 399.4±8.6 | 406.8±7.8 | 439.1±6.8 | 407.6±7.9 | 407.9±8 | 418.9±8.3[3] |
| Vitamin A density in RAE, µg/1000kcal | 484.1±14.1 | 385.2±18 | 293.8±12.3 | 265.3±8.7 | 478.9±14.1 | 369.7±18.8 | 308.4±11.2 | 244.6±8.4 |
| Vitamin D density, µg/1000kcal | 2.7±0.1 | 2.9±0.1 | 2.6±0.1 | 2.3±0.1 | 2.7±0.1 | 2.8±0.1 | 2.9±0.1 | 2±0.1 |
| Vitamin C density, mg/1000kcal | 87.2±3 | 54.1±1.7 | 39.6±1.9 | 35.1±1.6 | 80.7±3.1 | 55.1±1.9 | 38.8±1.8 | 36.2±1.7 |
| Sodium density, g/1000kcal | 1431±24 | 1441±22 | 1463±20 | 1474±19[9] | 1385±22 | 1436±23 | 1470±21 | 1530±20 |
| Naturally-occurring folate density, µg/1000kcal[12] | 163.3±3.8 | 119.1±2 | 94.7±1.8 | 91.5±2.2 | 160.7±3.9 | 115.3±2.1 | 99.1±2.1 | 85.7±1.9 |
| Folacin density from food sources, µg/1000kcal[13] | 214.6±3.6 | 180.3±2.6 | 159.9±2.6 | 151±2.4 | 214.6±3.7 | 173.7±2.5 | 158.6±2.7 | 150.7±2.4 |
| Phosphorus density, mg/1000kcal | 739.6±7.9 | 690.7±7.6 | 661.4±8.6 | 653.8±9.8 | 730.1±7.2 | 683.5±8 | 680±10.3 | 642.9±6.7 |
| Magnesium density, mg/1000kcal | 206.2±2.3 | 176.7±3 | 150.8±2 | 137.8±2 | 200.7±2.3 | 168.6±2.2 | 156.8±3.4 | 137±2.1 |
| Iron density, mg/1000kcal | 7.5±0.1 | 6.8±0.1 | 6.4±0.1 | 6±0.1 | 7.4±0.1 | 6.6±0.1 | 6.2±0.1 | 6.2±0.1 |
| Zinc density, mg/1000kcal | 6.1±0.1 | 5.5±0.1 | 5.4±0.1 | 5.3±0.1 | 6±0.1 | 5.5±0.1 | 5.4±0.1 | 5.3±0.1 |
| Potassium density, mg/1000kcal | 1855±19 | 1529±18 | 1294±14 | 1244±16 | 1768±19 | 1473±16 | 1380±20 | 1239±15 |
| Caffeine density, mg/1000kcal | 72.2±3.7 | 94±7.2 | 91.3±4.4 | 106.5±4.8 | 80.8±3.6 | 85.5±3.8 | 107.7±7.7 | 93.8±4.6[4] |
| Linoleic acid, % energy | 5.5±0.1 | 5.7±0.1 | 6±0.2 | 6.5±0.2[10] | 5.7±0.1 | 5.7±0.1 | 6.2±0.2 | 6.2±0.1[5] |
| Linolenic acid, % energy | 0.8±0 | 0.8±0 | 0.7±0 | 0.7±0[11] | 0.8±0 | 0.7±0 | 0.7±0 | 0.7±0[6] |
| Glycemic index | 27.3±0.5 | 32.8±0.8 | 34.8±1.9 | 29.9±0.6 | 26.9±0.4 | 29.6±0.5 | 37.3±1.9 | 30.8±0.6 |

CCHS: Canadian Community Health Survey; RAE: retinol activity equivalents; SDS: simplified dietary pattern score; wPLS: weighted partial least squares. Values are means ± standard errors. Estimates are weighted LS from a regression model adjusted for age, sex and energy misreporting status with bootstrapping to ensure accurate standard errors when using survey data. P-trends were estimated with the use of the ED, HF, LFD dietary pattern score or the simplified dietary pattern score in their continuous form and represents the P-value associated with the linear regression coefficient. All P-trends were <0.0001 unless otherwise noted.

[1] P-trend = 0.0208.

[2] P-trend = 0.0007.

[3] P-trend = 0.0043.

[4] P-trend = 0.0047.

[5] P-trend = 0.0003.

[6] P-trend = 0.0034.

[7] P-trend = 0.0085.

[9] P-trend = 0.0681.

[9] P-trend = 0.4329.

[10] P-trend = 0.0004.

[11] P-trend = 0.1226.

[12] Naturally-occurring folate includes various forms of folate found naturally in food.

[13] Sum of quantities of naturally occurring folate in addition to folic acid without considering their differing bioavailability.

A)

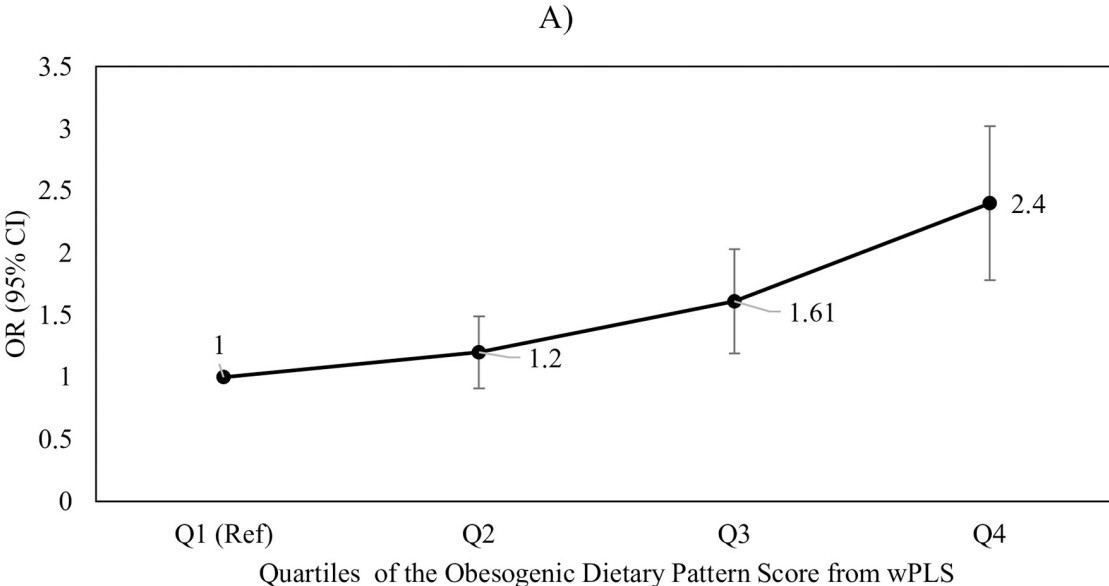

B)

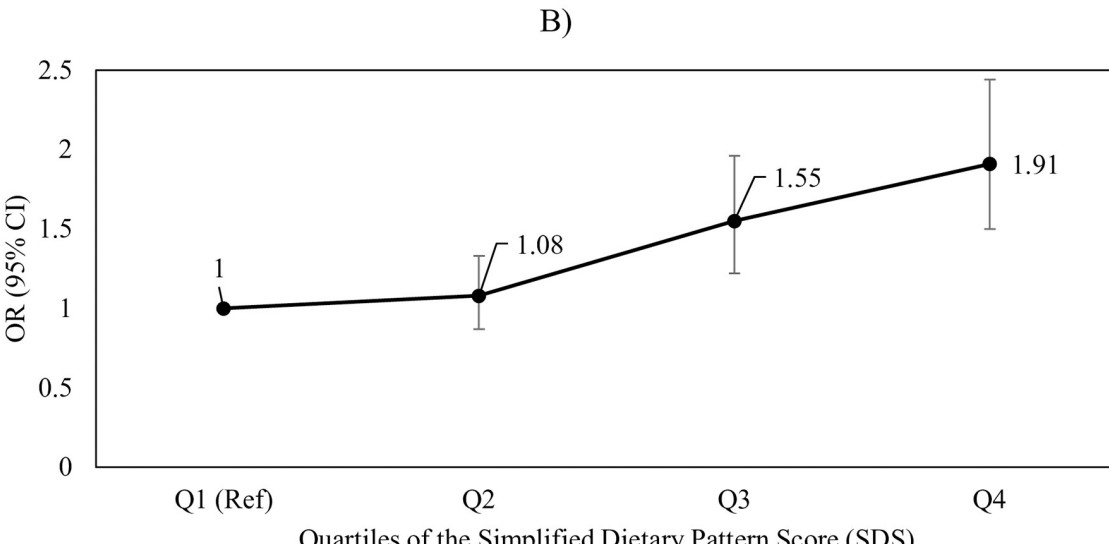

**Fig 2. Multivariate-adjusted odds ratios (ORs) and 95% confidence intervals (CIs) for the likelihood of being obese (BMI≥30) across quartile categories of A) simplified dietary pattern score (SDS) and B) the energy-dense, high-fat and low-fiber density ("obesogenic") dietary pattern score from the wPLS (weighted partial least squares) model among Canadian adults from CCHS-Nutrition 2015 (*n* = 12,049).** Q1 ("Healthiest") was used as reference. Both P-trends are <0.0001. Both models were adjusted for age, sex, physical activity, smoking status and energy misreporting status.

high fat and low fibre was consistently and strongly associated with increased odds of being obese, with or without accompanying chronic conditions. The findings from this study also confirm that hybrid dietary patterning methods and simplified diet pattern scoring techniques can perform just as well as *a priori*, index-based methods in tracking with obesity. Lastly, this study does not offer support for the "obesity phenotype" hypothesis [20], as there was no

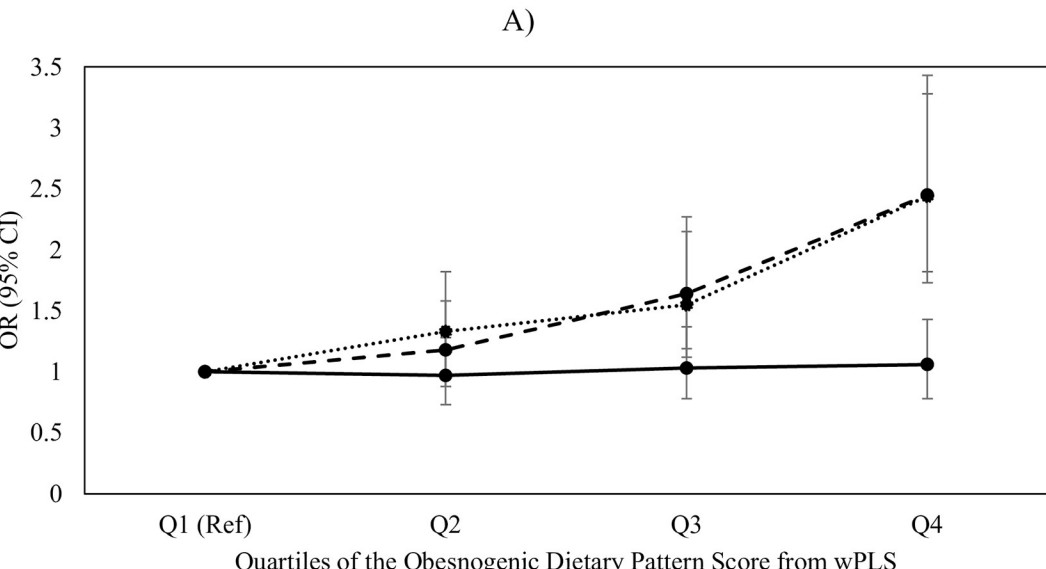

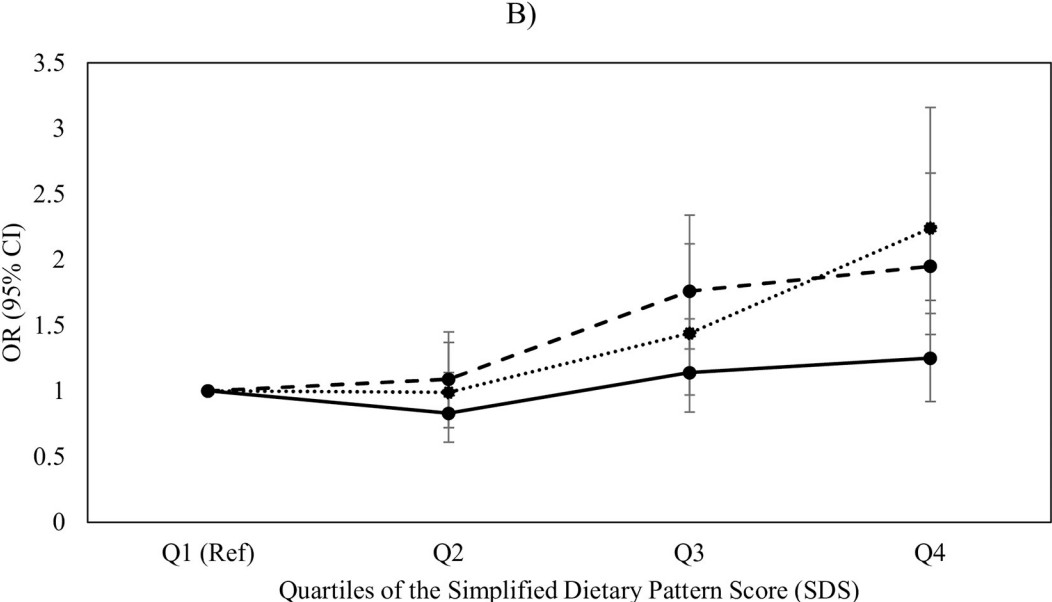

**Fig 3. Multivariate-adjusted odds ratios (ORs) and 95% confidence intervals (CIs) for the joint classification of likelihood of obesity with ≥1 self-reported chronic disease across quartile categories of A) the energy-dense, high-fat and low-fiber density ("obesogenic") dietary pattern score from the wPLS (weighted partial least squares) model and B) simplified dietary pattern score (SDS) among Canadian adults from CCHS-Nutrition 2015 (*n* = 12,049).** Normal weight (BMI<30 with ≥1 chronic conditions = solid line; "Healthy obesity" (BMI≥30 with no chronic conditions = dashed line; "Unhealthy obesity" (BMI≥30 with ≥1 chronic conditions) = dotted line. Q1 ("Healthiest") was used as reference. All models were adjusted for age, sex, physical activity, smoking status and energy misreporting status.

significant difference in likelihood of either healthy obesity or unhealthy obesity across quartiles of diet pattern scores in our sample.

Few studies have used hybrid methods to identify dietary patterns associated with body weight or obesity among adult populations [21–25]. A wPLS model was recently used to

identify an energy-dense, high fat and low fibre dietary pattern in a previous cycle of CCHS nutrition data, CCHS 2004 [5]. The results between CCHS 2004 and this current study are comparable: both wPLS and SDS scores were significantly associated with increased odds of obesity, with or without accompanying chronic conditions, and performed well compared to the 2015 DGAI.

Interestingly, the magnitude of some key food predictors in the wPLS model (i.e. those foods with factor loadings $\geq |0.17|$) have changed since 2004. Notable changes between 2015 and 2004 include: significant increased loadings for whole grains, legumes and soy, and salty snacks in 2015; a decrease in the factor loading for yogurt from >0.17 in 2004 to <0.17 in 2015; and less food predictors in the 2015 data have reached the significance cut-off. Some of these differences may be due to the fact that additional food categories were added as predictor variables in the CCHS 2015 wPLS model compared to 2004 to reflect new foods captured in CCHS 2015 (see S3 Table for more detail). Some of these changes may also simply reflect the dietary changes of Canadians since 2004. However, despite the magnitude of some predictor variables changing, the direction of the association has generally stayed the same–that is, no food that once loaded negatively has now loaded positively in 2015, or vice versa.

Notably, there are some findings in this study which have not changed since 2004. Certain foods have consistently (and insignificantly) loaded in the wPLS model across CCHS cycles, such as: tea, shellfish and vegetable oils in the negative direction; and eggs, whole-fat dairy products and alcohol in the positive direction (i.e. contributing to an obesogenic diet) [5]. These aforementioned foods and food groups have been inconsistently associated with either contributing to or being protective against obesity or weight gain in the scientific literature [26–32]. The fact that these foods did not load significantly in the wPLS model in either 2004 or 2015 cycles of CCHS questions the importance of focusing on these specific foods and food categories for NCD prevention.

The magnitude of the differences in food quantity across quartiles of wPLS and SDS scores compared to the relatively smaller differences in nutrient contribution of each respective diet pattern across quartiles (i.e. Table 4 in comparison to Table 5) also points to the importance of examining *food* patterns over nutrient patterns (along with foods with both positive and inverse associations with a disease-associated dietary pattern of interest) when examining relationships between nutrition and health. The findings from this particular study elucidates the top foods, food categories and beverages most associated with obesity among Canadian adults, which may be beneficial for more targeted public health policy for obesity prevention.

This study has its limitations. Only a single day of dietary recall was used for analyses, making the results prone to measurement error, which may reduce the precision and confidence in study findings. Additionally, BMI may not be an adequate marker of adiposity in this sample, as our sample includes older adults (who may lose height with age) or those with chronic conditions (who may lose weight due to their condition). However, as there are no additional markers of adiposity or obesity in CCHS, our use of BMI to examine obesity is justified. Further, the use of self-reported data to classify those with $\geq 1$ chronic conditions may have led to some misclassification of respondents, as individuals are likely to misreport chronic health conditions, and some conditions may go undiagnosed. And finally, this study utilizes cross-sectional data which is prone to residual confounding and reverse causation. This may partly explain null findings in regards to the "obesity phenotype" hypothesis, as those classified as unhealthy obese may have been living with obesity for a number of years, or may lose weight as their condition progresses [20]. We were therefore unable to tease apart the differential effect of diet on one's obesity *progression* in this study.

This study has multiple strengths. First, it utilizes the most-recent, nationally-representative data on Canadians' dietary intakes to characterize dietary patterns. Secondly, the use of

weighted partial least squares allows for a nuanced examination of a data-derived dietary pattern associated with obesity, and key food categories highly associated with such a pattern. Derivation of a simplified dietary pattern score also aids in the translatability of findings from this study to different datasets and sub-populations. Care was also taken to explore not only obesity, but obesity with and without accompanying disease, to further understand the relationship between diet, weight, chronic disease and health.

While estimates based on measured BMI from the 2017 Canadian Health Measures Survey pinpoint overweight and obesity rates at a standstill for the last decade [3], the burden of chronic diseases is still high [1, 2]. Evidence on the relationship between diet, obesity and chronic diseases will be paramount to aid in the development of specific, relevant and effective nutrition policies in the years to come. The results from this study pinpoint key foods consumed by Canadians that are associated with obesity and may help in the development of policies and dietary interventions for chronic disease prevention. That being said, more studies utilizing longitudinal data are required to substantiate study findings.

## Supporting information

**S1 Table. Scoring criteria for the Dietary Guidelines for Americans Adherence Index (DGAI) for individuals with 2000 kcal/day estimated energy requirement (EER).** (PDF)

**S2 Table. Percent variation explained from the wPLS-derived dietary pattern for two scenarios: Energy density, total fat intake and fiber density as response variables (Model A); and energy density, saturated fat intake and fiber density as response variables (Model B).** (PDF)

**S3 Table. Predictor variables used in weighted partial least squares (PLS) analysis in the Canadian Community Health Survey-Nutrition 2015.** (PDF)

**S4 Table. Odds ratios and 95% confidence intervals for the likelihood of "healthy" or "unhealthy" obesity, with normal weight (BMI<30) as reference.** Results are across quartiles of the energy-dense, high-fat and low-fiber density ("obesogenic") dietary pattern score from the wPLS (weighted partial least squares) model and B). (PDF)

**S5 Table. Odds ratios and 95% confidence intervals for the likelihood of "healthy" or "unhealthy" obesity, with normal weight (BMI<30) as reference.** Results are across quartiles of the simplified dietary pattern score (SDS). (PDF)

## Author Contributions

**Conceptualization:** Alena (Praneet) Ng, Mahsa Jessri.

**Data curation:** Alena (Praneet) Ng, Mahsa Jessri.

**Formal analysis:** Alena (Praneet) Ng, Mahsa Jessri.

**Investigation:** Alena (Praneet) Ng, Mahsa Jessri.

**Methodology:** Alena (Praneet) Ng, Mahsa Jessri.

**Supervision:** Mary R. L'Abbe.

**Validation:** Alena (Praneet) Ng, Mahsa Jessri.

**Visualization:** Alena (Praneet) Ng.

**Writing – original draft:** Alena (Praneet) Ng.

**Writing – review & editing:** Alena (Praneet) Ng, Mahsa Jessri, Mary R. L'Abbe.

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
