## [Decision Letter · Decision Letter 0]

9 Apr 2021

PONE-D-21-05281

Using partial least squares to identify a dietary pattern associated with obesity in a nationally-representative sample of Canadian adults: results from the Canadian Community Health Survey – Nutrition 2015

PLOS ONE

Dear Dr. Ng,

Thank you for submitting your manuscript to PLOS ONE. After careful consideration, we feel that it has merit but does not fully meet PLOS ONE’s publication criteria as it currently stands. Therefore, we invite you to submit a revised version of the manuscript that addresses the points raised during the review process.

We look forward to receiving your revised manuscript.

Kind regards,

David Meyre

Academic Editor

PLOS ONE

Journal Requirements:

Reviewers' comments:

Reviewer's Responses to Questions

**Comments to the Author**

1. Is the manuscript technically sound, and do the data support the conclusions?

Reviewer #1: Yes

Reviewer #2: No

2. Has the statistical analysis been performed appropriately and rigorously? 

Reviewer #1: I Don't Know

Reviewer #2: N/A

3. Have the authors made all data underlying the findings in their manuscript fully available?

Reviewer #1: Yes

Reviewer #2: Yes

4. Is the manuscript presented in an intelligible fashion and written in standard English?

Reviewer #1: Yes

Reviewer #2: Yes

5. Review Comments to the Author

Reviewer #1: Dietary patterns in industrialized countries are characterized by overconsumption of fats, sugars, and processed meats in the total caloric availability, leading to a rise of several non-communicable diseases, including obesity. Considering the complex challenge that represents dietary intake evaluation and its consequences, this study proposes an original way to study diet-disease relationships in a large sample of Canadian adults.

- Data from the Candian Health Survey Nutrition 2015 were used, providing a large and recent study population (20,487), but analyses were performed on a sample of 13,598 adults. The authors may justifiy why they didn’t use entire cohort. Also, It would be appreciable to describe the population briefly (age, sex…).

- The weighted partial least squares wPLS is a hybrid method previously used by Jessri et al. (2017) to derive an energy-dense, high-fat and low-fiber density dietary pattern. It would have been interesting to include high sugar intake. Indeed, the previously validated Dietary Guidelines for Americans Adherence Index (DGAI) is a diet quality index focused on overconsumption and energy density. Eleven index items assess the energy-specific food intake recommendations, including "Added sugar," and 9 assess the healthy choice nutrient recommendations. Dietary assessment is a classical one 24-hour recall. Even if all individuals were guided through the 24-hr dietary recall, it is well-known that total sugar intake is difficult to evaluate precisely.

- Then, the authors used a simplified diet score (SDS) constructed from the results of the wPLS regression using the methodology of Schulze et al., (2003). Thus, the authors chose a model based on energy density, total fat intake, and fiber density to define "obesogenic" dietary patterns, while excessive sugar intake would also be considered. Nevertheless, the authors assume study limitations (single day of dietary recall, BMI as obesity marker).

- This study represents a great update of the previous cycle of CCHS nutrition data (CCHS 2004) where wPLS was used, allowing comparison. In this way, despite the magnitude of some predictor variables changing, the association's direction has generally stayed the same. Interestingly, this article identifies some key food predictors that have changed since 2004 (an increase of whole grains, legumes, and soy & salty snacks and a decrease in yogurt). Others like tea, shellfish and vegetable oils or eggs, whole-fat dairy products, and alcohol have not changed since 2004 whereas these food groups have been inconsistently associated with either contributing to or being protective against obesity in the scientific literature.

Finally, even if the authors were unable to elucidate the differential effect of diet on one’s obesity progression in this study, this article contributes to evidence on the relationship between diet, obesity, and chronic diseases, an area that still lacks longitudinal data.

Reviewer #2: The outputs of this research published before.

Ng, A., Jessri, M., & L'Abbé, M. (2020). Identification of an Obesogenic Dietary Pattern Using Partial Least Squares in a Nationally-Representative Sample of Canadian Adults. Current Developments in Nutrition, 4(Supplement_2), 552-552

6. PLOS authors have the option to publish the peer review history of their article (what does this mean?). If published, this will include your full peer review and any attached files.

Reviewer #1: No

Reviewer #2: No

---

## [Author Response · Author response to Decision Letter 0]

13 Jul 2021

Responses to Reviewer #1:

Thank you for your comments. 

We aimed to analyze the dietary patterns of Canadian adults in this study; therefore, we removed children from the analyzes as their dietary habits are different from adults. The following changes have been made to Lines 114-116 for clarity:

“Weighted partial least squares analyses were originally performed on a sample of 13,598 adults to characterize the dietary patterns of the adult Canadian population.”

We have also included a brief description of the sample, in Lines 116-119:

“This sample of adults consisted of 49% males and 51% females. Of the sample, 16% were between 19-30y, 41% were between 31-50y, and 43% were above 50y. Based on measured BMI, 37% of the sample was normal weight, 36% was overweight, and 27% was classified as obese.”

While total sugar is indeed a nutrient of public health concern in Canada, the main aim of this study was to derive a dietary pattern using weighted PLS which would capture the World Health Organization’s classification of an “obesogenic diet”, i.e. one that is energy-dense, high in total fat, and low in fibre. Additionally, as we wanted to compare results between CCHS 2015 and CCHS 2004, we kept the model definitions the same as a previous study using CCHS 2004 data (see: Jessri et al. 2017. AJCN).

Reviewer #2: The outputs of this research published before.

Ng, A., Jessri, M., & L'Abbé, M. (2020). Identification of an Obesogenic Dietary Pattern Using Partial Least Squares in a Nationally-Representative Sample of Canadian Adults. Current Developments in Nutrition, 4(Supplement_2), 552-552

Responses to Reviewer # 2:

Thank you for your comment. A poster abstract of this study was previously presented at Nutrition 2020; all poster abstracts were published in Current Developments of Nutrition after the conference. The full manuscript has not been published.

---

## [Editor Report · Decision Letter 1]

16 Jul 2021

Using partial least squares to identify a dietary pattern associated with obesity in a nationally-representative sample of Canadian adults: results from the Canadian Community Health Survey – Nutrition 2015

PONE-D-21-05281R1

Dear Dr. L'Abbe,

We’re pleased to inform you that your manuscript has been judged scientifically suitable for publication and will be formally accepted for publication once it meets all outstanding technical requirements.

Kind regards,

David Meyre

Academic Editor

PLOS ONE
---

## [Editor Report · Acceptance letter]

28 Jul 2021

PONE-D-21-05281R1 

Using partial least squares to identify a dietary pattern associated with obesity in a nationally-representative sample of Canadian adults: results from the Canadian Community Health Survey – Nutrition 2015 

Dear Dr. L’Abbe:

I'm pleased to inform you that your manuscript has been deemed suitable for publication in PLOS ONE. Congratulations! Your manuscript is now with our production department. 

Kind regards, 

on behalf of

Dr. David Meyre 

Academic Editor

PLOS ONE